# Framing Slogans for Responsible Gambling Campaigns: A Tale of Two Models

**DOI:** 10.3390/healthcare11202754

**Published:** 2023-10-18

**Authors:** Lily Lim, Vincent Xian Wang

**Affiliations:** 1MPU-Bell Centre of English, Macao Polytechnic University, Macau SAR, China; 2Department of English, University of Macau, Macau SAR, China; vxwang@um.edu.mo

**Keywords:** persuasive message, responsible gambling, framing, Prospect Theory, Theory of Planned Behaviour, conceptual metaphor

## Abstract

This study investigates the persuasive mechanism of slogans employed in responsible gambling campaigns. We analyse slogans from official posters in the U.S., Singapore, and Macau, focusing on two domains. First, the Theory of Planned Behaviour is applied to examine the intention to gamble expressed in the slogans to reveal how gambling is positioned in social contexts. Second, two framing devices—i.e., conceptual metaphors and the frame of gains/losses—are examined to understand how these framing devices reinforce the persuasive message while interacting with each other. Two models of persuasion emerge from our data—one encouraged ‘grounded games’ for enjoyment, while the other discouraged gambling due to its potentially ‘harmful’ consequences. We advocate for a gestalt view on the theoretical constructs that contribute to the overall effectiveness of persuasive messaging. These constructs should be integrated into an analytical framework, with particular attention given to the framing effect of conceptual metaphors and the gain/loss frame, and their interplay.

## 1. Introduction

Commercial gambling has evolved into a “global business” in the 21st century, with its worldwide gross revenue doubling from 2000 to 2019 [1]. This robust growth stems from its legalisation by governments across a wide range of countries and jurisdictions in the past decades—such as the U.S., the U.K., Australia, New Zealand, Canada, France, Germany, Spain, Italy, Norway, the Czech Republic, Poland, Russia, Ukraine, South Africa, Brazil, Colombia, Singapore, Malaysia, Japan, Vietnam, and India [2,3,4]. The rapid spread has been further fuelled by technological advancements such as modern electronic gambling machines [3,5], mobile sports betting [2,3], and easily accessible online games [6,7]. However, gambling has also brought about significant social impacts, and its potential harm to citizens has caused deep concern. Research has revealed that the negative impacts of gambling can include financial difficulties and bankruptcy, family conflict and violence, disruption of relationships, individual anxiety, mental disorders, health issues, substance abuse, criminal activities, and suicidality [2,3,8,9,10,11].

At this juncture, responsible gambling (RG) has emerged as a pivotal concept, forming the basis for the development of measures, strategies, and policies aimed at preventing and reducing harm related to gambling [7,10,11,12,13]. The importance of RG is underscored particularly in its role in safeguarding vulnerable cohorts of the population [14] and maintaining social stability [1,15], as exemplified in the widely influential Reno model [16]. Over the past few decades, responsible gambling has been systematically, periodically, and vigorously promoted by both the central and state governments, as well as by the gambling industry, across the globe, leading to its rapid growth as a research area [5,6,7,9,10,11,12,13,14,15,16,17,18,19,20]. 

In RG campaigns, slogans prominently displayed on posters and banners in public spaces play a key role in conveying messages about healthy gambling to audiences [5,21]. These slogans are designed for effective communication and encapsulate the various conceptualisations of problem gambling. Although, to date, there are only a few studies that specifically investigate slogans used in RG campaigns [5,20], research on RG has extensively focused on understanding gamblers’ perceptions, in order to develop effective interventions for guidance and persuasion. This stream of research tends to employ the Theory of Planned Behaviour (TPB) [22,23] as the dominant theoretical framework, conducting survey questionnaires that provided evidence supporting TPB’s prediction that individuals’ intention to gamble is strongly correlated with their engagement in gambling activities [15,18,24,25,26,27,28,29,30,31,32]. While TPB has been effectively applied to probe gamblers’ perceptions and predict their behaviour, it has not been utilised yet to investigate the appeal of RG messages designed to persuade gamblers to alter their behaviour. This area holds considerable potential, since research evidence has underscored the value of TPB in evaluating the efficacy of behaviour-changing messaging [33,34]. In our investigation, we found TPB particularly valuable for examining slogans in terms of their ways to position gambling and direct gamblers’ intentions. This analysis involves scrutinising the underlying constructs of intention to act—namely, attitude, subject norms, and perceived behaviour control (cf. Section 3.1).

Aside from the stance and positioning expressed by the slogans, the framing of a message can significantly enhance its appeal and persuasiveness. Based on the existing literature, we examine two notable framing devices—i.e., conceptual metaphors [35] and the prospect of gains/losses [36]. Although there is a substantial body of literature on conceptual metaphors for health communication [37,38,39] and an extensive body of work on the framework of gains or losses in health messaging and gambling studies [40,41,42,43,44,45,46,47,48,49], remarkably, there are still few studies that investigate the combined effect of these two types of framing [50]. 

Our literature survey suggests that neither type of framing has garnered significant attention in RG research. Only a few studies have investigated the use of metaphors in the context of gambling. One noteworthy metaphor, housework, was proposed as a valuable tool for effectively communicating the responsibility for addressing health concerns, including the reduction of gambling-related harm, within the Maori community in New Zealand [51]. In addition, a recent study examined Australian gamblers’ use of tropes in their discourse, which uncovered gamblers’ beliefs about who they perceived was responsible for achieving and reinforcing RG [9]. Moreover, we identified two studies that explored the use of metaphors to promote gambling activities. One study investigated the framing effect of pilgrimage as a meaningful metaphor, enabling outbound Chinese gambling tourists to interpret their experiences [52]. The other study analysed commercial advertisements for online betting, identifying the metaphors ‘the act of love’, ‘a market’, ‘a sport’, and ‘a natural environment’ used to illustrate and promote betting activities [53]. From our literature survey, it is clear that to date, very little is known about metaphorical framing in the gambling domain in general and in RG messaging in particular. Given the widely acknowledged role of metaphors in shaping individuals’ perceptions and communication—as expounded by Lakoff and Johnson in their influential book Metaphors We Live By [35]—research on RG messaging and metaphor usage is urgently needed. 

This research collected data from the U.S., Singapore, and Macau to unpack the theoretical constructs underlying persuasive messaging. The three locations merit special attention in their own right. First, the U.S. commercial gambling industry generated a record revenue of USD 60 billion in 2022 [54]. While the majority of states have legalised casino gambling and sports betting [2,14], other forms of gambling activities such as online gambling, lotteries, horse racing, and fantasy sports are also widespread across the country [4,55,56]. Large-scale epidemiological studies indicated that the prevalence of lifetime pathological gambling in the U.S. adult population fluctuated between 0.4% and 0.6% [11]. In response, RG is consistently and regularly advocated by the National Council on Problem Gambling (https://www.ncpgambling.org/programs-resources/responsible-gambling/, accessed on 13 October 2023) and gambling operators. 

Second, Macau has a long history of legalised gambling dating back to 1847 [57]. It surpassed the Las Vegas Strip in gambling revenue in 2006, becoming the world’s biggest gambling centre [58]. Casino gambling is the economic pillar of the city, and it remains the only city in China where casinos are legal. The prevalence of gambling disorder in Macau is estimated to be between 0.8% and 2.1% [59], with associated issues identified including depression, thoughts or attempts of suicide, family problems, and alcohol and drug abuse [60]. The local government has maintained a steadfast commitment to public education on RG and coordinates annual RG promotional events [60].

Finally, in 2005, the Singapore government decided to construct two casino resorts to enhance its tourism industry, in line with its self-styled image as the ‘Renaissance city of Asia’, supplementing traditional lotteries and horse racing [3]. The National Council on Problem Gambling (NCPG) in Singapore conducts surveys triennially, with the most recent report for the year 2020 indicating that probable pathological gamblers constituted approximately 0.02% to 0.4% of the respondents [61]. Both problem gamblers and their family members have been able to benefit from preventive and protective services [62]. For instance, as of March 2017, a total of 285,024 legal orders had been issued barring individuals from entering local casinos (including self-exclusion orders). Moreover, between July 2015 and June 2016, there were 20,748 calls made to the helpline [63]. NCPG not only introduced the helpline system, but also strives to disseminate RG messages in collaboration with gambling operators [63].

This research undertakes an analysis of slogans for RG messaging employed in the three locations. We are interested in examining both the stance on gambling expressed in the slogans and the framing devices used to further reinforce that stance. Our research questions are the following:(a)What stance is expressed in the slogans concerning the positioning of gambling in relation to players’ financial well-being and social lives?(b)What conceptual metaphors are employed in the slogans, and how do they contribute to the message to gamble responsibly?(c)Is the frame of gains or the frame of losses conveyed in the slogans, and how do these frames contribute to the message to gamble responsibly?

## 2. Materials and Methods

### 2.1. Materials

In order to collect slogans from the three target locations, we first conducted image searches via Google Chrome, using the keywords ‘responsible gambling’ and ‘responsible gaming’ to yield a range of 200–300 image snapshots, incorporating location-specific words such as ‘Singapore’, ‘the U.S.’, and ‘Macau/Macao’ as necessary. We also conducted searches via other web browsers such as Firefox and Microsoft Edge and found that the search results were very comparable with those generated by Google Chrome. 

By manually browsing through the images and single-clicking on the relevant ones to display similar images, we collected images that feature slogans specific to the three locations. Our selection criteria prioritised slogans that verbally articulate the concept of responsible gambling, with a particular emphasis on those that employ a catchy phrase designed to resonate with—and impart a particular message to—the audience. We therefore excluded a substantial number of images that lacked a specific slogan. This category includes, for instance, posters—predominantly from various U.S. states—that simply announce events such as “Responsible Gambling Week” or “Problem Gambling Awareness Month”, with specific event dates appended.

From our searches, we also identified the key institutions that appeared in the images—for example, the American Gaming Association (AGA, https://www.americangaming.org/, accessed on 13 October 2023), the Singapore National Council on Problem Gambling (NCPG, https://www.ncpg.org.sg/, accessed on 13 October 2023), and the Direcção de Inspecção e Coordenação de Jogos (DICJ, or ‘Gaming Inspection and Coordination Bureau’ in English, https://www.dicj.gov.mo/web/en/responsible/responsible01/content.html, accessed on 13 October 2023). We accessed the websites of these institutions to retrieve their slogans and posters. 

In addition, entities within the gambling and entertainment industry, such as the New York Lottery (https://nylottery.ny.gov/, accessed on 13 October 2023) and Resorts World Sentosa Singapore (https://www.rwsentosa.com/en, accessed on 13 October 2023), also contributed slogans and posters. In Macau, major gambling corporations regularly participate in the promotion of RG events coordinated by the Macau government and also provide relevant educational materials through kiosks installed within casinos. However, the design of the most widely circulated and noticeable annual campaign slogan and poster in Macau is undertaken by the Institute for the Study of Commercial Gaming. We retrieved the slogans for Macau from this source, covering a period of more than ten years. Our data from the U.S. and Singapore primarily encompass the past three years. 

Data collection was conducted in the first half of 2023. The consolidated collection comprises 18 slogans and associated posters from the U.S., 11 from Singapore, and 12 from Macau. We also obtained permission to reproduce representative posters from the producers of the promotional materials.

### 2.2. Methods

The slogans were analysed in two main domains. First, we applied the Theory of Planned Behaviour (TPB) [22,23] to examine how the intention of gambling is construed in the slogans. Specifically, we looked at the attitudes, subjective norms, and perceived behavioural control that underpin individuals’ intention to gamble [23,25]. The aim was to reveal how gambling is positioned in relevant social contexts, as expressed by RG campaigners. 

Second, we investigated how the message expressed by the campaigners’ stance on gambling is further enhanced by framing devices, including conceptual metaphors and the frame of gains/losses. We employed the Metaphor Identification Procedure VU University Amsterdam (MIPVU) [64] to examine lexical units in context to determine cross-domain mapping and drew on the Conceptual Mapping Model proposed by Ahrens [65] to explore the specific elements within the source domain that were mapped onto the target domain—namely, the ‘Mapping Principle’ [65,66]. Conceptual metaphors are widely recognised for their persuasive power, while the frame of gains/losses, developed by Tversky and Kahneman’s Prospect Theory [36], has been extensively investigated in health communication [47]. We are particularly interested in the interplay between these two framing devices for persuasive messaging, and this contributes to the theoretical innovation of our study. Our research on the integration of the two framing devices also echoes recent studies on framing for effective persuasion [50,67].

## 3. Results

Table 1 presents representative slogans collected from the U.S., Singapore, and Macau (cf. Section 2.1). A significant number of the slogans use plain language and do not incorporate a metaphor. Conversely, a myriad of slogans used across the three locations employ metaphors of construction, driving, war, and even the strategy of playing all-in (in a poker game) to elucidate the concept of responsible gambling, while the metaphors of tornado and child’s play are utilised to demonstrate what gambling is and is not, respectively. 

The slogans and their associated posters vary not only in their use of metaphor but also in the way in which they promote RG. One subset of slogans portrays positive scenarios, characterised by excitement, smiling faces, and cheerful body language to affirm that gambling can be enjoyable when the players maintain control. By contrast, another subset presents images of sad faces, desperate body language, and a gloomy environment to discourage individuals from engaging in gambling activities.

We therefore identified two main methods of promoting RG in our data—i.e., Models A and B in Table 1—which emerged as patterns of persuasive messaging in the slogans used in the three places. The first model depicts gambling in a positive light—for example, the slogans “Keep your game grounded” (New York Lottery) and “Are you playing smart?” (Singapore Responsible Gambling Forum)—and posits that gambling is a leisure activity for the individuals who can manage it. The second model casts a negative light on gambling—for example, the slogans used in Macau that compare gambling with a tornado or a house that loses its balance—to warn gamblers that they can lose control and eventually lose the most valuable things in their lives including their families. Model A is used relatively more frequently than Model B, as indicated in Table 1. We then proceeded with more in-depth qualitative studies on the two models.

We investigated the two models that promote RG in terms of (a) their construal of the intention of gambling, using the conceptual framework of the Theory of Planned Behaviour (TPB) [23], and (b) the framing of messages using conceptual metaphors and the prospect of gains or losses for enhancing persuasiveness. Our analysis of the intention of gambling aims to reveal the slogans’ stance in terms of the positioning of gambling in people’s conceptual and social worlds, while our examination of the framing devices focuses on how the positioning of gambling is further sharpened through the use of framing devices to convey more persuasive messages.

### 3.1. Intention of Gambling

In this section, we apply TPB to examine how gambling is portrayed in the two models of persuasion by slogans. In the light of TPB, people’s gambling behaviours are closely related to their intention of gambling, and the latter is a strong predictor of the former. TPB proposes that people’s intention to carry out a certain activity mainly relies on three factors—i.e., their attitude towards the behaviour, the subjective norms about the performance of the behaviour and its acceptability, and their perceived behaviour control over the activity in which they are engaged. In the following sections, we investigate the intention of gambling and its three constructs, which are expressed in the two models for RG promotion (Section 3.1.1 and Section 3.1.2).

#### 3.1.1. Model A: Grounded Games for Sustainability 

The first model demonstrates a favourable attitude towards gambling. Of the three constructs of attitude, the attitude on gambling expressed in this model is largely positive. The model portrays gambling as a form of entertainment, often light-hearted enjoyment. It does not present gambling as a means of making money, though, or as an exciting adventure in search of good fortune. For example, in the 2019 version of the “Are you playing smart” slogan presented by the Responsible Gambling Forum (RGF) in Singapore (Figure 1), a well-dressed smart-looking young man displays his budget plan, in which 5% of his income is allocated to “leisure, entertainment, lottery etc.”. The idea here is that gambling is about one’s expenditure, rather than a means for gains, which should be duly budgeted for, along with other expenses such as schooling costs, savings and mortgages. The key to RG is that one should allot a reasonable portion of their expenditure to gambling and maintain control of that budget. This model appears appropriate for the situation in Singapore (In Singapore, responsible gambling materials are specifically made visible to gamblers rather than the general public. These materials are not broadcast in the mainstream media and are only available in gambling venues). According to the National Council on Problem Gambling (NCPG) surveys [61], in 2020, 44% of the population took part in various betting activities, with a median monthly betting amount of SGD 15. The estimated pathological and problem gambling rate remained low at around 1%. The U.S. slogan of “Keep your game grounded” expresses a similar light-hearted optimism about gambling, promoting the view that keeping control of your gambling is key to doing it right and remaining healthy (Figure 2). 

Second, the subjective norms expressed in this model strongly suggest that gambling is acceptable to the general public and it is fine to engage in it, as long as it is under control. This model treats gambling as something morally permissible rather than controversial, and this position is observed, for example, in the “grounded” and “playing smart” slogans. Similarly, in the slogan “set limits to control your spend, deposits, and the time spent wagering” (cf. Table 1), which was developed by Draft Kings and the American Gaming Association, wagering is compared to driving a vehicle. The comparison implies that betting is part of life, as is driving, but one needs to set limits to remain in control.

Third, the perceived control over gambling is emphasised particularly strongly in this model of “grounded games”. The type of control explicitly referred to here is not about gamblers’ control over the outcome of gambling though—such as through an individual’s gambling skills or competence, let alone through the superstitious behaviours practised by some gamblers to manipulate the results of games. The scope of control promoted in this model is about individuals’ ability to keep their gambling activities grounded without exceeding the limit they have set on their spending. For example, the “setting limits for spending” slogan (cf. Table 1) features a mechanical speedometer displayed in the middle of the graphic design, photographically drawing on people’s driving experience to illustrate their need to control their gambling spending. Specific strategies and tactics for control have also been formulated and communicated to the public. For example, in a 2022 poster published in Singapore (Figure 3), the “playing smart” slogan is followed by eight principles that gamblers should follow to stay in control, such as “set a limit in advance and keep to it”, “don’t chase your losses”, and “don’t gamble when you are distressed or upset”. The efforts aim at equipping gamblers with both strategic insights and practical tactics to be in control and keep their gambling grounded. 

In summary, this model of persuasion promotes “grounded games” and conveys the message that people’s intention to gamble is acceptable, and this is in line with the social norms of gambling by the general public. This model does not seek to dissuade people from betting but focuses on conveying the message that individuals should exercise self-control in order to act responsibly and make their betting enjoyable, reasonable, and sustainable.

#### 3.1.2. Model B: Gambling Can Get Disastrously out of Control

Unlike Model A, this model aims to frustrate people’s intention of gambling. We again apply the TPB framework to examine the constructs of intention, and observe that, first of all, the attitude towards gambling is depicted in a strongly negative light and is even stigmatised at times. The negative attitude derives from the gloomy prospect depicted in the slogans that gamblers will lose their money, car, home, and so on rather than making money, and there is therefore no reason to want to gamble. For reasonable individuals, the loss of valuable belongings and close family relationships would be disheartening and distressing, akin to being ravaged by a powerful tornado (cf. Figure 4, left). Consequently, vigilance and diligence are needed to guard against such circumstances. By instilling a negative image into the mind of gamblers, this model works to reduce or even eliminate the desire to gamble.

In addition, the subjective norms invoked in this model indicate that gambling is abnormal, harmful, and hazardous, entirely at odds with maintaining a functional, caring, and united family. It is considered to be dangerously disruptive, as illustrated by the 2022 poster with the “lost control” slogan developed by the Institute for the Study of Commercial Gaming (ISCG) in Macau (Figure 4, right), in which a home built on sports betting and casino games is losing its balance and about to topple over. The norms and conventions invoked in this model of persuasion strongly disapprove of people engaging in gambling.

Finally, in terms of perceived behaviour control, this model shows little confidence in gamblers’ ability to exercise self-control. As demonstrated in the slogan from Macau (Figure 4, left), gamblers’ self-control is doomed to failure in facing the allure of gaming, as powerless as individuals’ efforts to prevail against a tornado. The “lost control” (Figure 4, right) slogan suggests that gamblers can eventually succumb to uncontrollable gambling, totally giving in to the allure of betting. The slogan attempts to convey the message that only by realising that one’s family and other valuables are about to be lost can gamblers have a chance to turn away from the games. 

Therefore, the second model of persuasion for RG endeavours to thwart gamblers’ intention to gamble by vigorously highlighting the disastrous consequences of gambling. In terms of attitudes, subjective norms, and perceived behaviour control in relation to gambling, the model works to reduce the ground that supports each of the three constructs of the intention of gambling. Model B employs a relatively more aggressive approach to address the problem of pathological gambling than Model A. 

### 3.2. The Framing of RG Messages

In this section, we examine how persuasive messages expressed in Models A and B are further framed using two types of framing devices—(a) conceptual metaphor and (b) the frame of gains and losses—to enhance their appeal. 

#### 3.2.1. Model A: The Framing of Grounded Games 

This model positions gambling as a pleasant and socially and ethically permissible activity that can be enjoyed by gamers with self-control. Our analysis of the slogans and the graphic designs of the posters associated with this model reveals the use of conceptual metaphors to reinforce the message. For example, the slogan “Keep your game grounded” (Figure 2) uses the metaphor of construction, or more specifically, the building of a sturdy and stable structure. The metaphor selectively borrows the characteristic of a firmly grounded building from the source domain of construction and applies this quality to advocate the optimal method of gambling in the target domain. The mapping principle can be postulated as follows: engaging in gambling can be understood as constructing a building, in that a soundly grounded building is firm, secure, and robust and so is sensible gambling practice. This concept is also communicated through the design of the animation.

The web-based version of the promotional material uses video animation that shows letters in robust fonts joining together to form words such as “keep”, “your”, “game”… and the words stacking together to construct the phrase “keep your game grounded”. The phrase is visually presented in a solid structure that is firmly grounded (see Figure 2). The animation shows a leisurely sunny day with birds chirping and people of different ethnicities greeting each other. The construction metaphor is therefore apt to convey the message that only when people make their gambling well-grounded can they achieve pleasant social lives and build a harmonious community.

Aside from the use of metaphors, the persuasiveness of the message is further enhanced by the frame of gains. According to Prospect Theory [36], people tend to be risk-averting when they perceive themselves as situated in a gain frame. As demonstrated by Figure 2, the design of the promotional material gives rise to a conspicuous gain frame—things are going well for everyone on a pleasant sunny day—and the theoretical prediction is that they will tend to avoid risks so that they can continue to enjoy a happy and balanced life. Risk-averting behaviours would include gambling responsibly and being vigilant about avoiding unaffordable bets (that is, serious risks). The frame of gains therefore potentially encourages people’s tendency to gamble responsibly, from the perspective of Prospect Theory.

In Model A, the integration of metaphors and the incorporation of a frame of gains synergistically convey the message that gambling should be approached with a solid grounding, much like a well-constructed building. The frame of gains further cultivates a tendency for risk aversion, potentially fostering RG behaviours that enable players to maintain enjoyable, well-balanced, and sustainable lives. 

#### 3.2.2. Model B: Gambling Can Disastrously Get out of Control

The second model of persuasion portrays gambling in a negative light, both morally and financially. Conceptual metaphors serve as powerful tools to illustrate this perspective. For example, metaphors such as a tornado or a toppling building (cf. Figure 4, left and right) vividly highlight the detrimental effects of betting, which can have devastating consequences for gamblers’ families and finances, causing painful and often irreparable damage. It is worth noting that the metaphor of construction is utilised in both Models A (cf. Section 3.2.1) and B, although with distinct mapping principles. In Model B’s scenario featuring a toppling building, the metaphor of construction selectively focuses on a building with unstable foundations to depict the precarious nature of pathological gambling and the inevitable outcome of loss and harm. The mapping principle can be articulated as follows: problem gambling can be understood as constructing a building on shaky ground, in that such a building is destined to collapse just as pathological gambling ultimately leads to a loss of balance and irreparable damage. The objective of using conceptual metaphors to underscore negative aspects is to motivate gamblers to guard against potential losses and improve their self-control.

However, the emphasis on heavy losses in this model could potentially undermine the goal of promoting RG, from the prospect-theoretic perspective. Prospect Theory predicts that people tend to be risk-taking when they perceive themselves as being in a losing situation. In particular, gamblers who are already struggling with addiction may relate to the bleak scenarios highlighted in the posters and perceive themselves as being in a distressingly bad situation. Consequently, some gamblers may resort to risk-taking behaviours, believing that the only way to recover from their troubled situation is by achieving significant gains through betting, despite the high risks involved. 

That said, we are not arguing that the depiction of heavy losses will foster risk-taking behaviours in every gambler. What Prospect Theory proposes is that individuals, based on a rational assessment of risks, are more likely to take risks when the same situation is presented to them in a loss frame than in a gain frame. Therefore, rational calculation of the risks involved still forms the basis of individuals’ decision making. However, from the prospect-theoretic perspective, it is more effective in promoting RG if the dangers are expressed in a gain frame rather than in a loss frame. For example, stating that “Exercising self-control in gambling allows you to avoid losing all your money, your house and your family” would be more effective than asserting “You will lose all your money, your house and your family if you continue gambling”, because the former utilises a gain frame that more effectively fosters individuals’ risk-averting behaviour than a loss frame would. 

In summary, Model B of persuasion utilises metaphors of natural and man-made disasters to demonstrate the devastating effects of gambling. However, the frame of losses may potentially encourage risk-taking behaviours in some problem gamblers. This point deserves particular attention in future studies from researchers, campaigners, and administrators in the field.

## 4. Discussion

This research has identified two representative models employed in RG campaigns. The main findings are summarised in Table 2, and we seek to offer a more comprehensive understanding of the theoretical constructs that contribute to the persuasiveness of the models. However, determining which model is more effective is beyond the scope of this study. There is a real need in future research to empirically investigate the relative effectiveness of each model for different cohorts of gamblers and to determine which of the theoretical factors outlined in Table 2 resonate more tangibly with specific groups of gamblers. 

Model A avoids dissuading individuals from engaging in gambling activities. Instead, it sees gambling as an admissible form of recreation, emphasising the power of effective management to ensure players’ enjoyment while safeguarding their security, both financially and in terms of their overall well-being. This philosophy is reinforced by the metaphor of a firmly grounded building, highlighting the need for a stable and controlled approach to gambling. The gain frame advocated by Model A further fosters risk-averse behaviours among gamblers, aiming to protect them from potential harms. 

By contrast, Model B actively discourages gamblers from engaging in gambling activities, portraying them as harmful and morally wrong pursuits with serious financial consequences, as well as family breakdown and other negative outcomes. This discouraging perspective is reinforced by the metaphor of a tornado and the vivid depiction of a building on the brink of collapse, symbolising the inherent risks of gambling. However, a potential drawback of Model B lies in its impact on individuals who are already grappling with gambling problems. The model’s approach may amplify a frame of losses, potentially driving gamblers towards further risky behaviours in an attempt to alleviate their troubled situation, leading to further adverse outcomes. Given the real-life application of Model B, it is crucial to investigate whether problem gamblers are more prone to develop increased risk-taking behaviour as a result. Future empirical studies can provide valuable insights into the potential implications of Model B’s discouraging message on problem gamblers’ behaviour. 

The two models can be examined in the context of the ongoing debate in which the dominant Reno model was challenged and a public-health-focused approach was proposed to replace it [68]. A school of researchers stressed the need to safeguard public health and public interest against the harm of gambling, while also questioning the effectiveness of RG measures initiated by the gambling industry itself [1,68]. They critiqued the prevailing discourse on RG, challenging the validity of solely relying on individual gamblers’ self-control to resolve the problem of pathological gambling [69]. In relation to this scholarly debate, Model A aligns more closely with the Reno model, which centres on the concept of RG that depends on individuals’ exercise of self-control. By contrast, Model B delves into the harm of gambling and exposes the negative consequences on gamblers’ finance, health, and family relationships as its persuasion strategy. This embodies a public-health-centred approach to harm prevention, contrasting with the Reno model. The two models therefore largely represent the two major scholarly positions on gambling. 

We see that both an individual’s self-control and external factors such as legislation, entry restrictions, and non-proximity to casinos are important factors contributing to his/her engagement with gambling. There is a pressing need for empirical evidence; however, our literature survey identified only one empirical study. This study found a limited impact of a statewide advertising campaign on problem gambling, revealing a low exposure rate of 8% among adult residents of Indiana [21]. Aside from large-scale surveys, empirical case studies could also provide valuable insights in future research, particularly concerning negatively framed messaging related to the public-health-centred model. For example, the message “Gambling is a family disease. One person may be addicted but the whole family suffers.” (https://cdn4.geckoandfly.com/wp-content/uploads/2018/10/anti-gambling-gambler-quotes-09.jpg, accessed on 13 October 2023) frames gambling as a disease and underscores the collective suffering of the gambler’s family. Gathering evidence on how individuals perceive the constructs and framing features of such messages would significantly enhance our understanding.

We advocate for a gestalt view on persuasive messaging, emphasising the significance of theoretical developments in the growing field of RG research. There have been a considerable number of survey studies on gambling that have applied the Theory of Planned Behaviours (TPB) to examine gamblers’ intentions and empirically test the correlations between TPB parameters. Notably, strong correlations have been found between gamblers’ gambling practice and their intentions, and also between problem gamblers’ perceived behavioural control and their level of engagement in gambling [15,18,24,25,26,27,28,29,30,31,32]. In RG campaigns, persuasive messaging needs to articulate these TPB theoretical constructs to capture gamblers’ attention and reshape their gambling intentions, ultimately having a positive influence over their gambling behaviour. The patterns or models formed by the theoretical constructs demonstrated in this study warrant thorough consideration from campaigners, researchers and the gambling industry.

Since the slogans collected in this study had a limited scope and were exploratory in nature, it is important to note that its findings on the models of persuasion and the attitudes towards gambling should not be viewed as representative of broader trends or characteristics of the locations where the slogans were produced and used. In fact, we can observe that different models of persuasion are often employed concurrently by various campaigners promoting RG in the same area. For example, in Singapore, the Responsible Gambling Forum (RGF), a partner of the National Council on Problem Gambling (NCPG), utilised the slogan ‘playing smart’, which exemplifies Model A and emphasises sustainable gambling practices. On the other hand, NCPG has released more than ten advertisements (https://www.ncpg.org.sg/resources/advertisements, accessed on 13 October 2023) that vividly depict the challenging circumstances of problem gamblers and seek to provide them with channels for assistance, reflecting the use of Model B, which portrays the severe consequences of uncontrolled gambling. NCPG endeavours to maintain a high level of vigilance regarding problem gambling and has implemented a comprehensive system to monitor and reduce visits of problem gamblers to betting venues (https://www.ncpg.org.sg/services/overview-of-exclusions-and-visit-limit, accessed on 13 October 2023), showcasing a tangible response to pathological gambling identified in Model B, and employing a robust and effective system. Given their significant practical value in safeguarding problem gamblers, the functionality of response systems to problem gambling deserves focused research attention. 

## 5. Conclusions

This research has investigated the theoretical constructs underlying the messages conveyed by slogans that promote RG. Two representative models have been identified—one encouraging ‘grounded’ gambling with self-control, and the other dissuading players from gambling by highlighting its devastating ‘harms’. The two models demonstrate sharply different stances on gambling in terms of players’ financial well-being, social lives, and self-control. Each model utilises unique conceptual metaphors or distinct mapping principles to strengthen its message, while the models further contrast with each other in terms of the frame of gains/losses expressed. 

The first model fosters gamblers’ risk-averting behaviour to maintain enjoyable, sustainable, and recreational gambling by exercising self-control, promoting smartness, reason, and a well-managed life. By contrast, the second model appeals to gamblers’ sense of responsibility towards their families and their desire to protect themselves from grave financial losses due to gambling. However, the frame of losses expressed in the second model may potentially provoke risk-taking behaviours in problem gamblers, according to Prospect Theory [36]. There is therefore an urgent need for further empirical studies to test the persuasive appeal of both models, and to determine their strengths and drawbacks in providing effective messaging, especially in relation to specific cohorts, such as pathological gamblers. 

## Figures and Tables

**Figure 1 healthcare-11-02754-f001:**
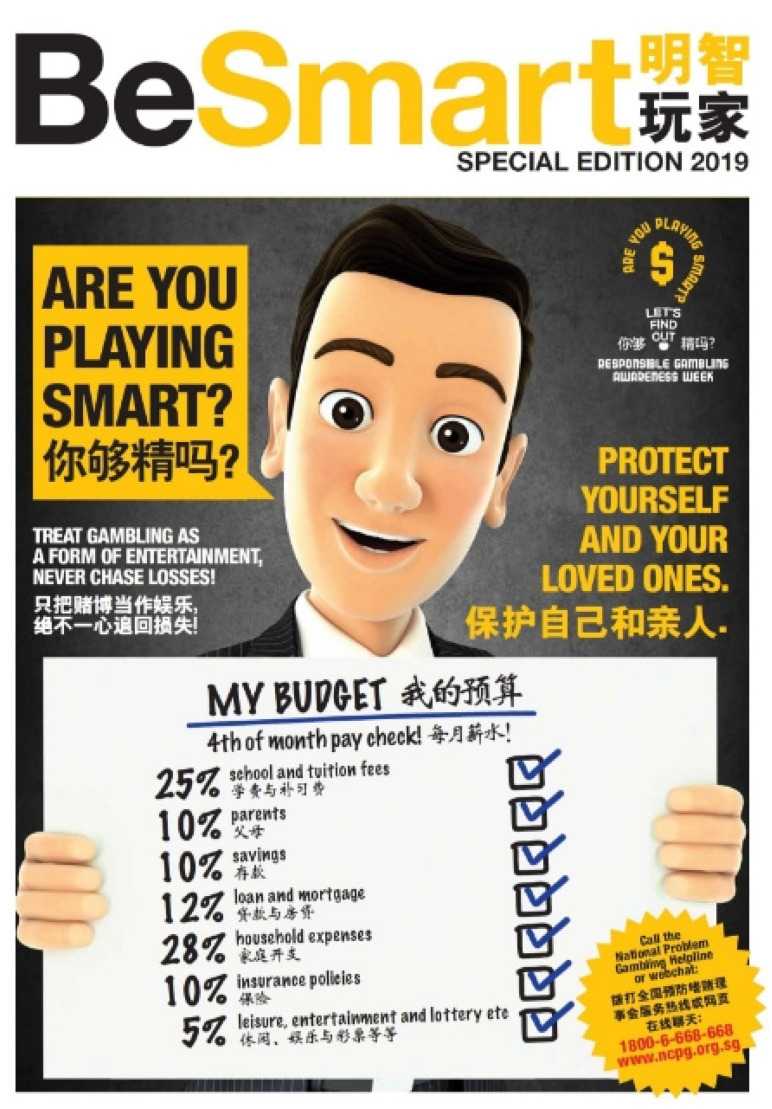
A poster for responsible gambling in Singapore: 2019 by RGF.

**Figure 2 healthcare-11-02754-f002:**
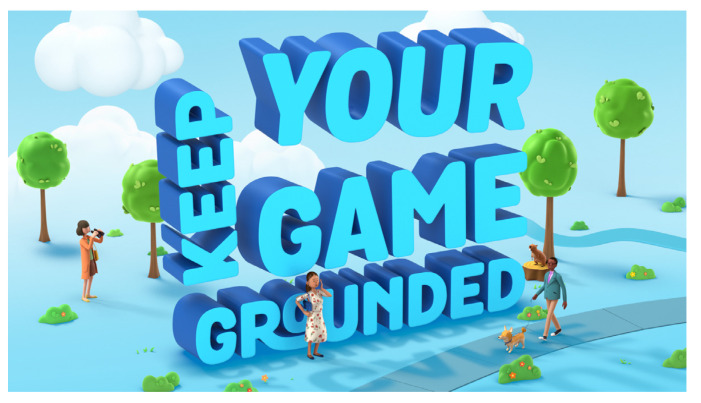
A poster for responsible gambling in the U.S.: 2021 by NY Lottery.

**Figure 3 healthcare-11-02754-f003:**
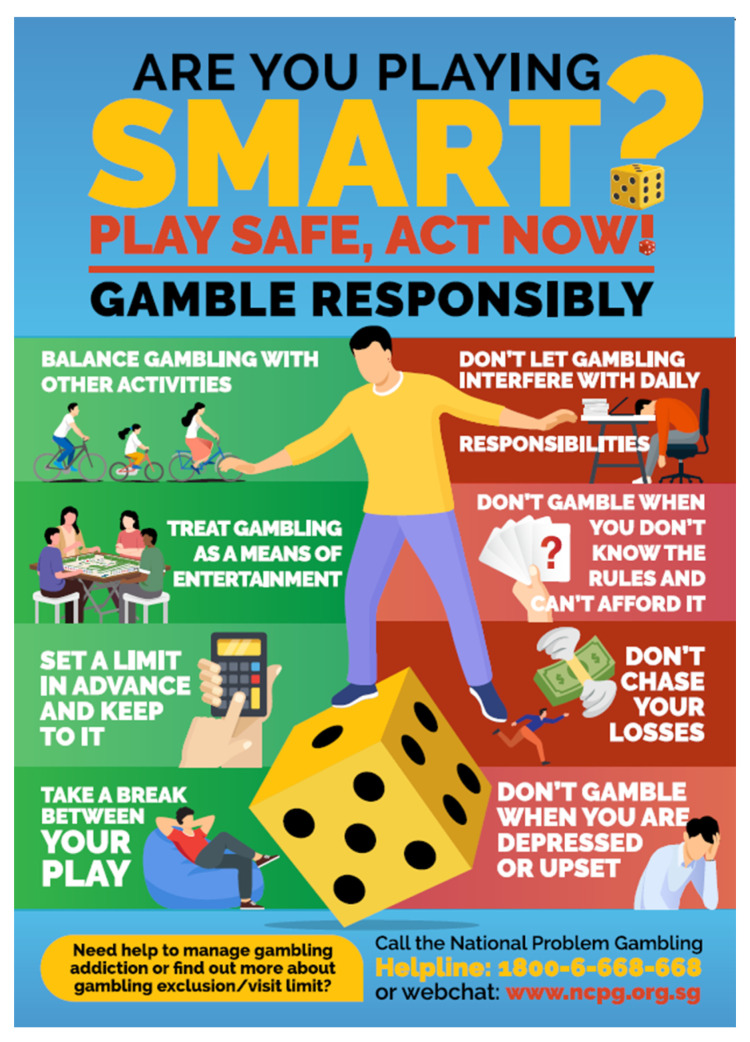
A poster for responsible gambling in Singapore: 2022 by RGF.

**Figure 4 healthcare-11-02754-f004:**
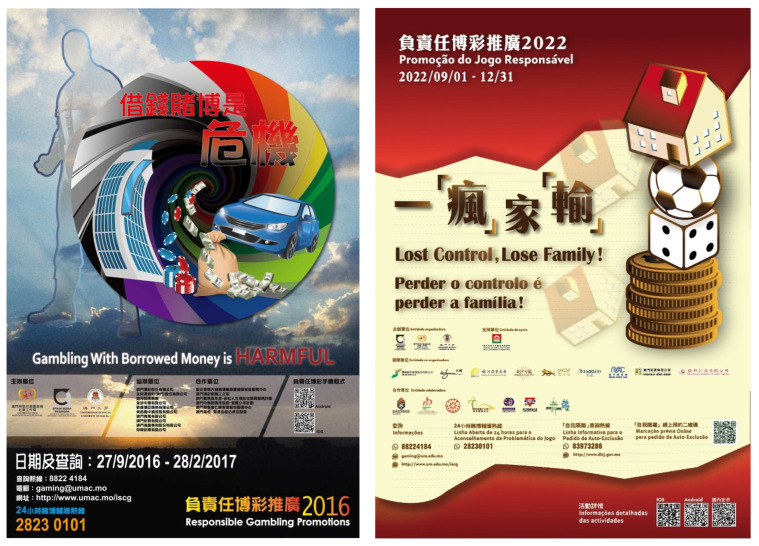
Posters for responsible gambling in Macau: 2016 and 2022 by ISCG.

**Table 1 healthcare-11-02754-t001:** Selected slogans from the U.S., Singapore, and Macau: Models A and B.

	Metaphor	Model	URL	Campaigner
U.S.				
Keep your game grounded	construction	A	https://vimeo.com/531964137, accessed on 13 October 2023	New York Lottery
Limits: set limits to control your spend, deposits, and time spent wagering	driving (also image)	A	https://www.3blmedia.com/news/research-roundup-responsible-gaming-tool-usage-and-positive-play, accessed on 13 October 2023	American Gaming Association (AGA); Draft Kings
Take your gambling in a new direction	driving (also image)	A	https://www.problemgambling.ie/uploads/9/0/0/2/9002949/gear-workbook.pdf, accessed on 13 October 2023	Oregon GEAR program
Remember: Lottery tickets are not child’s play	child’s play	A	https://gamblinghelp.org/, accessed on 13 October 2023	Florida Council on Compulsive Gambling
Time to go all-in on responsible gambling	going all-in (in a poker game)	B	https://www.playca.com/5511963/responsible-gambling-importance-california/, accessed on 13 October 2023	Play CA (California)
Gamble responsibly!	n/a	B	https://smartcasinoguide.com/responsible-gambling/, accessed on 13 October 2023	Smart Casino Guide
Have a game plan Stick to a budget	n/a	A	https://haveagameplan.org/, accessed on 13 October 2023	AGA
If you gamble, Get set before you bet If you gamble, Be the 95%	n/a	A	https://www.facebook.com/photo/?fbid=553128246975557&set=ecnf.100068351598829, accessed on 13 October 2023	New Hampshire Council on Problem Gambling
It’s important to know when to stop	n/a	A	https://www.swmbh.org/members/gambling/, accessed on 13 October 2023	Southwest Michigan Behavioral Health
Singapore				
Overcome problem gambling, seek help today	war	A	https://www.ncpg.org.sg/resources/ncpg-brochures, accessed on 13 October 2023	National Council on Problem Gambling (NCPG)
You have a say Protect your family	war	A	https://www.ncpg.org.sg/resources/ncpg-brochures, accessed on 13 October 2023	NCPG
Protect yourself and your loved ones	war	A	https://www.rwsentosa.com/en, accessed on 13 October 2023	Resorts World Sentosa Singapore
Are you playing smart?	n/a	A	https://www.turfclub.com.sg/en/responsible-gambling.html, accessed on 13 October 2023	Singapore Turf Club
Gamble responsibly	n/a	A	https://www.turfclub.com.sg/en/responsible-gambling.html, accessed on 13 October 2023	Resorts World Sentosa Singapore; NCPG
Self-help is your best bet	n/a	A	https://www.ncpg.org.sg/resources/ncpg-brochures, accessed on 13 October 2023	NCPG
Be a winner, Be a smart player	n/a	A	https://www.gentingrewards.com.sg/en/home/casino/responsible-gambling/before-gambling, accessed on 13 October 2023	Resorts World Sentosa Singapore
Macau				
Lost [sic] control, Lose family	construction (in image)	B	https://www.gov.mo/zh-hant/news/931539/, accessed on 13 October 2023	Direcção de Inspecção e Coordenação de Jogos (DICJ), 2022
Gambling with borrowed money is harmful	tornado (in image)	B	https://www.um.edu.mo/iscg/img/rgposter/2016.png, accessed on 13 October 2023	DICJ, Responsible Gaming Promotions 2016
Gambling is not business, stay in control!	business	A	https://www.um.edu.mo/iscg/img/rgposter/2019.png, accessed on 13 October 2023	DICJ, Responsible Gaming Promotions 2019
Good people get wealthy by good means	n/a	A	https://www.um.edu.mo/iscg/img/rgposter/2011.jpg, accessed on 13 October 2023	DICJ, Responsible Gaming Promotions 2011

**Table 2 healthcare-11-02754-t002:** Two models of persuasion for RG: a summary.

	Model A	Model B
Examples of slogans	‘grounded’, ‘playing smart’	‘harmful’ (tornado), ‘lost control’
Intention of gambling	Permissible as leisure	Strongly discouraged
Attitude	Positive	Negative
Subjective norms	Permissible	Disapproval
Perceived behavioural control	Achievable	Futile
Metaphors	Firmly grounded construction driving	Natural and man-made disasters, toppling building
Prospect Theory		
Frame of gains or losses	Frame of gains	Frame of losses

## Data Availability

The slogans and the corresponding posters are publicly available on the web.

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
