# Peer review of "Framing Slogans for Responsible Gambling Campaigns: A Tale of Two Models"

_healthcare, 2023, doi:10.3390/healthcare11202754_

Round 1
Reviewer 1 Report
This paper provides an interesting analysis of how slogans related to gambling are framed as either a gain or loss and the metaphors associated with each frame.
The study currently provides a qualitative analysis of posters about gambling from the US, Singapore and Macau. The study would benefit from a clearer presentation of where the posters were found (i.e., websites), the dates they were presented to the public and accessed, and then a count of how many posters were analyzed from each country/time period, and what the criteria were for inclusion on the corpus.
In addition, in the analysis of the data, it should be made clear how many of the posters from the whole corpora were found to have attributes of either frame (as well as how many were found to involve metaphors in the slogans). The paper should also mention that the focus is on verbal and not visual analysis.
Additionally, the paper lacks from setting the ground with a straightforward quantative analysis before moving to a qualitative analysis (which is quite well done). The paper also needs to cite relevant work in the literature about how the source domain is determined for the metaphors that are proposed.
In addition, it is interesting to note that it appears that the two different frames involve different mapping principles (Ahrens, 2010) for the same source domain of building ("firmly ground construction" versus "toppling building"). Of course, in order to look at mapping principles, the ground first needs to be laid with a clear explanation of the how the corpus was created, the size of the corpus, and a clear quantitative analysis of what was found in the corpus. To this end, putting the slogans and their location/access date/publication date, metaphors used, frames associated with the slogans on Open Science Framework (or the equivalent) would be extremely helpful for the reviewer and for the reader to understand the full scope of the data analyzed.
Author Response
We are very thankful for your comments and suggestions, and have made revisions accordingly:
- In Section 2.1 (Materials), we have detailed the corpus of the slogans, specifically regarding data collection and the contents of the corpus.
- A new Table 1 has been included to feature representative slogans and outline the main trends of the data. We have added two paragraphs at the beginning of Section 3 (Results) to elaborate on Table 1.
- We have incorporated literature on metaphor identification and Mapping Principles in Section 2.2 (Methods).
- In Sections 3.2.1 and 3.2.2, we have examined the Mapping Principles for the use of the construction metaphor in the two distinct models.
- We have appended a paragraph to the end of Section 4 (Discussion) to address limitations and future studies.
Thank you again for your time and recommendations.
Reviewer 2 Report
This study focuses on different campaigns addressing responsible gambling. The idea of comparing responsibility campaigns in different jurisdictions is interesting, but the data includes only five posters. I suggest that the authors would look for more posters and read more about responsible gambling. It is also important to inform the international readers about the gambling culture in each jurisdiction. The dichotomy of good and bad gambling is too obvious: the authors should study more carefully the persons, colors, metaphors and texts in each poster/campaign. The authors could also discuss what is adequate and what is missing in these campaigns.
Reviewer 3 Report
The authors have addressed an important aspect of modern culture. The topic is relevant in the context of campaign policy for preventing pathological gambling. The authors have analyzed persuasive factors of two models of slogans. The author can improve their manuscripts by adding sections like strengths and limitations.
The English is overall good in this manuscript. However, professional proof-reading service will increase the fluidity of the language remarkably.
Author Response
We are very grateful for your comments and suggestions, and have made revisions to the article:
- We have added a paragraph to the end of Section 4 (Discussion) to address limitations and future studies.
- The paper now has been edited by an experienced copy editor.
- In Section 1 (Introduction), we have summarised the harmful consequences of irresponsible gambling, drawing from relevant literature, and expanded the discussion of previous studies related to our research.
- We have expanded the discussion of previous studies related to our research in Section 1 (Introduction).
- In Section 2.1 (Materials), we have detailed the corpus of the slogans, specifically regarding data collection and the contents of the corpus.
- A new Table 1 has been included to feature representative slogans and outline the main trends of the data. We have added two paragraphs at the beginning of Section 3 (Results) to elaborate on Table 1.
- We have incorporated literature on metaphor identification and Mapping Principles in Section 2.2 (Methods), and in Sections 3.2.1 and 3.2.2, we have examined the Mapping Principles for the use of the construction metaphor in the two distinct models.
Thank you again for your time and recommendations.
Reviewer 4 Report
The authors examine slogans used on posters located in the United States, Singapore, and Macao, with the aim of discovering the theoretical constructs that contribute to persuasive messages related to gambling. They apply the Theory of Planned Behavior (TPB) to examine how intention to gamble is interpreted, as well as conceptual metaphors and the win/loss framework to investigate how the message expressed by activists' stance on gambling is reinforced by framing devices. They found two models of persuasion, the first posits the idea that gambling is an enjoyable activity for people who can handle it, while the second warns players that they can lose control and eventually lose the most valuable things in their lives.
My comments and suggestions are:
It is suggested to expand the works related to the applied methods.
It is suggested to include more information about the problem of gambling to emphasize the importance of the study, e.g. number of people affected, among others.
It is suggested to make a comparison with similar studies, if any.
In figures x and y, add numbering and their respective reference in the text.
